# Economic burden and cost-effectiveness of treatments for open tibia fractures in Malawi: Economic analysis of a multicentre prospective cohort study

Alexander Thomas Schade[1,2,3*], Linda Alinafe Sande[1], Ewan Tomeny[2],
Maureen Sabawo[4], Nyamulani Nohakhelha[3], Kaweme Mwafulirwa[3],
Leonard Banza Ngoie[5], Andrew John Metcalfe[6], David Griffith Lalloo[2],
William James Harrison[7,8], Jason J. Madan[6], Peter MacPherson[9,10]

**1** Malawi-Liverpool-Wellcome Trust, Blantyre, Malawi, **2** Liverpool School of Tropical Medicine, Liverpool, United Kingdom, **3** Queen Elizabeth Central Hospital, Blantyre, Malawi, **4** Kamuzu University of Health Sciences, Blantyre, Malawi, **5** Kamuzu Central Hospital, Lilongwe, Malawi, **6** University of Warwick Medical School, Coventry, United Kingdom, **7** Countess of Chester NHS Foundation Trust, Chester, United Kingdom, **8** AO Alliance, Davos, Switzerland, **9** University of Glasgow, Glasgow, United Kingdom, **10** London School of Hygiene and Tropical Medicine, London, United Kingdom

* alexander.schade@gmail.com

## Abstract

### Background

Open tibia fractures result in substantial lifelong disability for patients, and are expensive to treat. As the injury typically affects young working men, the societal costs from open tibia fractures are likely to also be high in low income countries, but remain largely unknown. We therefore investigated the overall societal costs and cost-effectiveness of different orthopaedic treatments at one year following an open tibia fracture in Malawi.

### Methods

This study was a cost-utility analysis nested in a prospective cohort study from the healthcare- and societal-payer perspectives with a one-year time horizon. We obtained quality-adjusted life years (QALYs) from the EuroQoL 5 Dimension 3 Level (EQ-5D-3L) and patient lost productivity estimates at 6 weeks, and 3, 6, and 12 months post-injury. QALYs were calculated from utility scores were modelled within a hierarchical Bayesian multivariate modelling framework that jointly estimated individual-level trajectories in EQ-5D-3L scores and costs over follow-up. Direct treatment costs were obtained from a micro-costing study, and staff interviews at tertiary and district hospitals. Cost-effectiveness was reported in terms of societal cost per quality-adjusted life year (QALY). All costs were reported in 2021 United States dollars (USD).

**Data availability statement:** All relevant data can be found via the OSF repository using the following URL: https://osf.io/8fz6x.

**Funding:** This research was funded in whole by Wellcome [Grant number 203919]. For the purpose of open access, the author has applied a CC BY public copyright licence to any Author Accepted Manuscript version arising from this submission. The funders had no role in study design, data collection and analysis, decision to publish, or preparation of the manuscript.

**Competing interests:** "The authors have declared that no competing interests exist".

## Results

Between February 2021 and March 2022, 287 participants with open tibia fractures were included. There were substantial costs to participants one year following injury with 42% (n = 112) working with a median monthly household income of US$40 (IQR: US$7−90) compared to 89% (n = 255) working pre-injury, with a median monthly household income of US$60 (IQR: US$36−144). The posterior median of societal costs at one year varied between US$751 (80% credible intervals [CrIs]: US$-751−2,389) for treatment with plaster of Paris (POP) in a district hospital for a Gustilo III injury, to US$2,428 (80% CrIs: US$995−5027) for intramedullary nail in central hospital for a Gustilo III injury. The largest cost-effectiveness from a societal perspective was between an intramedullary nail and amputation for a Gustilo III injury with a posterior mean of US$2,290 (95%HDI: 36−4,547) per QALY.

## Conclusion

The main finding was that open tibia fractures result in significant costs to patients, the healthcare system and society in Malawi. Although the funding of orthopaedic treatment can be difficult in countries with very limited healthcare budgets, the costs to society of ignoring this issue are very high. A re-balancing of health budgets (including from government and donors) is needed to prioritise trauma care to reduce the growing societal economic burden from injury.

## Background

Globally, road traffic injuries are predicted to cost the world economy US$7·86 trillion between 2015–30, which is equivalent to an annual loss of 0.12% in global gross domestic product [1]. This includes an estimated US$180 billion likely to be lost annually due to injury in low and middle-income countries (LMICs) [2], higher than the WHO estimates for cancer ($100 billion per year), respiratory disease ($106 billion) or diabetes ($28 billion) in LMICs [3]. Road traffic injuries are in the top ten causes for Disability-adjusted life years (DALYs) in Malawi in 2021 [4].

Open tibia fractures (where the tibia is fractured and protrudes through the skin) are common following road traffic injuries and are a major driver of household poverty as they are often extremely debilitating and typically affect young working people, especially men [5]. The Lancet Commission on Global Surgery has prioritized protection against catastrophic expenditure following fractures as an important measure of financial access to essential surgical and anaesthesia care [6]. In high-income countries, open tibia fractures are expensive to treat, and only 60% of participants fully return to work at one year after injury [7,8]. In low-income countries, hospitals have fewer resources [9] and the impact on patients is likely to be even more severe, but has not been previously investigated [8].

Malawi is one of the poorest countries in the world (GDP per capita of $396 in 2021 [10]) and the economy largely depends on substantial multilateral economic

assistance from the International Monetary Fund, the World Bank, and bilateral assistance from donor nations; an estimated 80% of healthcare treatment in Malawi is funded from international donors [11]. Injury in global health still disproportionately lacks investment in trauma care as it received less than 1.0% of all disease specific global health assistance, while HIV/AIDs, tuberculosis and malaria received 36.0% of disease specific global health assistance in 2017 [12]. The large majority of the Malawi population is rural, but urbanisation is rising and, coupled with the increased availability of motor vehicles – particularly motorbikes – is contributing to the increasing burden of injury in cities [13]. The public health system is government funded, but has one of the lowest per capita expenditures on health in Africa and a total expenditure on health of 9.8% of GDP (far less than the 15% WHO recommendations [14,15]). The health system consists of three levels of healthcare facilities: primary care is provided in health centres; fracture care is provided at secondary care in district hospitals (typically rural); and tertiary care in referral hospitals (typically urban). Most fractures in Malawi are treated non-operatively by non-physicians called Clinical Officers [16]; national guidelines recommend that open tibia fractures should receive antibiotics, debridement, definitive fixation, and soft tissue coverage [17].

Economic evaluations of health interventions are important to inform policy decisions for healthcare funding and resource allocation. A systematic review of reported costs for open tibia fractures suggested that the initial hospitalisation costs varied widely in terms of measurement and setting and almost half of the studies (47%, n = 16/34) were conducted in the United States [8]. Therefore, we aimed to investigate the economic burden of open tibia fractures from a patient, healthcare and societal perspective and describe the cost-effectiveness of different orthopaedic treatments in Malawi to help inform policymakers.

## Methods

### Setting and population

This was an economic analysis that was part of a national multicentre prospective cohort study conducted in Malawi. The study took place at six hospitals, including two central hospitals (Queen Elizabeth Central Hospital [QECH] and Kamuzu Central Hospital [KCH]) and four district hospitals (Dedza District Hospital, Ntcheu District Hospital, Balaka District Hospital, and Machinga District Hospital). The study protocol was previously published in 2021 [18]. The participants included adults (18 years or older) who had open tibia fractures classified as Arbeitsgemeinschaft für Osteosynthesefrage (AO) Foundation/Orthopaedic Trauma Association class 42 [19] between 12th February 2021 and 15th March 2022. Individuals who were unable to provide consent or complete patient-reported outcome questionnaires were excluded from the study.

### Procedures

#### District hospitals

Each of the 28 districts in Malawi has its own government district hospital (each with approximately 140 beds and a catchment area of ~714,000 people on average). All study participants admitted to one of the study district hospitals received intravenous ceftriaxone, analgesia and temporary immobilisation, and were entirely managed by orthopaedic Clinical Officers working for the public health system. Patients were managed with irrigation and debridement under local anaesthetic in a minor theatre. Patients that needed operative fixation (intramedullary nail or external fixation) or soft tissue procedures (flap or skin graft) were transferred after debridement to tertiary hospitals. After wound review at 48 hours post-operatively, a plaster technician would apply full cast above knee plaster with a window to monitor the wound. This would be changed at 6 weeks follow-up.

#### Tertiary hospitals

Each of the four regions in Malawi has one tertiary referral government hospital (with an estimated 1,000 beds and a catchment population of five million people on average). All study participants admitted to a tertiary hospital received

intravenous antibiotics, analgesia and temporary immobilisation in the emergency department [20]. They were reviewed by the orthopaedic team on call and transferred to the orthopaedic ward. All patients were managed with irrigation and debridement under spinal/general anaesthesia. During admission, the patient had a daily junior doctor review and a senior consultant review once a week. Definitive operative treatment occurred a median 4 days (interquartile range (IQR): 1–9) after injury; fractures were managed with plaster of Paris, intramedullary nail, external fixation, or amputation based on clinical decision and theatre availability. All patients would buy crutches before discharge, as these were not provided by the hospital. All patients were advised to keep non-weight-bearing for 6 weeks.

### Effectiveness: Health-related quality of life (EQ-5D-3L)

We measured the effectiveness of each treatment strategy (amputation, external fixation, intramedullary nail, POP in central hospitals and POP in district hospitals) using quality-adjusted life years (QALYs) based on the EQ-5D-3L [21]. At recruitment, participants were asked to retrospectively complete the questionnaire for their health state pre-injury and at each subsequent follow-up visit, study. Research Assistants administered the EQ-5D-3L to the study participants. The EQ-5D-3L is a tool used to measure health-related quality of life (HRQoL) that has been translated to Chichewa (the local language) and validated for use in Malawian orthopaedic patients [22]. Utility scores were calculated from EQ-5D-3L responses using the Zimbabwean tariff, as no Malawi set exists [23]. QALYs were calculated from the utility scores using the area under the curve (AUC) method [24]. We calculated QALYs for each treatment split by fracture Gustilo grade, a grading system for open fracture severity, with Gustilo I the least severe, and Gustilo III the most severe fracture type [25]. We included the standard and common treatment modalities for each Gustilo grade. For Gustilo I and II, we calculated the healthcare costs for POP in the central hospital, POP in the district hospital and intramedullary nailing, whereas for Gustilo III, we calculated for amputation, external fixator, and intramedullary nailing. The presence of a fracture-related infection [26] was verified by examining the medical records for confirmatory clinical signs including purulent discharge, the presence of sinus/fistula, or wound breakdown. In cases where the patient couldn't attend the clinic, they were asked about the presence of purulent discharge during a telephone interview or home tracing visit.

### Costing

Direct medical and overhead costs, as well as indirect patient costs, were calculated. For treatments performed in central hospitals, we used micro-costing methods [27] to quantify the direct and indirect hospital costs associated with different open tibia treatments stratified by fracture Gustilo I/II and Gustilo III grade. Each procedure was observed between 1–3 times. For treatment in district hospitals, interviews were conducted with hospital staff (orthopaedic clinical officers, anaesthetic clinic officers, ward managers, plaster technicians and the District Health Officer) to estimate the direct and indirect hospital costs associated with orthopaedic treatment strategies.

Direct medical costs included: procedure personnel salaries and supplies; medications and investigations; surgical implants; and instruments, walking aids, extra investigations or medications associated with hospital stay and follow-up visits. No discounting was performed as follow-up was at most one year after injury. Outpatient costs included clinic personnel and radiography costs. Any participant that was readmitted had details of reason, length of stay and operations recorded. Direct non-medical costs included patient transportation and food whilst away from home. Participants reported household income and working status at baseline and each follow-up visit. Indirect costs included loss of productivity which was estimated by calculating the difference between pre-injury household income for a year and the reported household income at each follow-up.

### Cost-effectiveness

Cost-utilities were calculated by dividing the healthcare and societal costs by the QALY of each orthopaedic treatment. QALYs are calculated by multiplying the duration of time spent in a health state by the utility score (EQ-5D) associated

with that health state, whereas DALYs are calculated using age-weighting, discounting of future DALY losses and broad survey methods for calculating disability weights [28]. Currently, DALY weights do not exist for open tibia fracture in Malawi [29]. DALYs and QALYs willingness to pay are similar for interventions and recent work suggests that the willingness to pay in Malawi might be as low as US$61 per DALY averted [30,31].

## Statistical analysis

To investigate trajectories in household income and the percentage of households reporting an income of US$0 per month, we constructed a hierarchical Bayesian multivariable model with inference drawn using Markov chain Monte Carlo (MCMC) sampling. Models were fit using the R `brms` package as an interface to CmdStanR in R [32]. We modelled household income and the percentage of households reporting an income of $0 per month using a hurdle gamma distribution, and included participant-level random intercepts to account for autocorrelation. We drew 4000 samples for each parameter from posterior distributions summarised the household income and percentage of participants reporting an income of US$0 pre-injury up to one-year post-injury using means and quantile-based uncertainty intervals. Separately, we modelled the percentage of participants who reported being in work at each time point using a binomial distribution.

We additionally constructed Bayesian multivariate regression models to jointly estimate healthcare costs, societal costs, and HRQoL, accounting for within-participant correlation, allowing us to predict trajectories in cost-utility following orthopaedic interventions. Because treatments were guided by setting (district or central hospital) and injury severity (Gustilo grade), we fitted models for participants with Gustilo I/II and Gustilo III fractures separately. Initial hospital costs were included at the closest follow-up (6 weeks). Readmission costs were added to the closest follow-up visit. The difference between one year societal costs and HRQoL for different interventions was calculated. Code and data are available at (https://doi.org/10.17605/OSF.IO/N36EG).

## Ethics statement

The study was approved by the College of Medicine Research and Ethics Committee (COMREC Ref number: P.09/20/3130) in Malawi, and the Liverpool School of Tropical Medicine Research Ethics Committee (Reference number: 20–068). Written informed consent was obtained from all participants.

## Results

We recruited 287 participants including 224 participants from tertiary hospitals and 63 participants from district hospitals. Most participants in tertiary and district hospitals (255/287, 89%) reported being in work prior to injury, with 59% (170/287) reporting their occupation as "casual labour/business". Overall, the pre-injury median monthly household income was US$60 (IQR: US$36–144) and one year post-injury median monthly household income was US$40 (IQR: US$7–90). Only 42% (n = 112) were working at one year after injury.

We included data from 253 participants who completed one-year follow-up, including 140 participants with Gustilo I or II injuries, and 113 participants with Gustilo III injuries; the main reasons for non-inclusion in outcome analysis were: non-standard, rare treatment modalities or missing hospital length of stay. Cases involving amputation (n = 5), plates (n = 2) and external fixation (n = 17) for Gustilo I or II injuries and cases involving plates (n = 3) for grade III fractures were excluded in the models. Seven cases were missing length of hospital stay. Empirical healthcare costs and societal costs are shown in Table 1. In summary, healthcare costs at one year post injury ranged from US$86 (IQR: 84–104) for plaster of Paris (POP) in a district hospital for a Gustilo I or II injury to US$348 (285–435) for an intramedullary nail in a central hospital for a Gustilo III injury. Median empirical hospital costs for participants with fracture-related infections were higher at one year (median: US$494, IQR:368–626) than those with no fracture-related infection (median: US$345; IQR:159–415). Median empirical hospital costs increased with the severity of injury: Gustilo I (median: US$196, IQR: 105–411), Gustilo II (US$363, IQR:192–459), Gustilo III (US$404, IQR:317–548).

**Table 1. Empirical patient and hospital costs (2021 USD).**

| Patient costs | Central Hospital (n = 224) | | | | | District hospital (n = 63) | | TOTAL |
|---|---|---|---|---|---|---|---|---|
| Occupation: | | | | | | | | |
| • Casual labour/business | 136 (61%) | | | | | 34 (54%) | | 170 (59%) |
| • Farmer | 22 (10%) | | | | | 17 (27%) | | 39 (14%) |
| • Corporate employee | 26 (12%) | | | | | 3 (5%) | | 29 (10%) |
| • Driver | 11 (5%) | | | | | 08 (13%) | | 11 (4%) |
| • Student/unemployed | 24 (11%) | | | | | 1 (2%) | | 32 (11%) |
| • other | 5 (2%) | | | | | | | 6 (2%) |
| Health insurance | 9 (4%) | | | | | 1 (2%) | | 10 (4%) |
| Poorest Quintile | 175 (78%) | | | | | 63 (100%) | | 238 (83%) |
| Median monthly Household costs (2021 USD) | | | | | | | | |
| Baseline | 72 (36-144) | | | | | 60 (24-120) | | 60 (36-144) |
| 6 weeks | 9 (0-60) | | | | | 15 (0-60) | | 12 (0-60) |
| 3 months | 14 (0-61) | | | | | 0 (0-60) | | 12 (0-60) |
| 6 months | 30 (0-72) | | | | | 12 (0-60) | | 24 (0-60) |
| 1 year | 48 (9-84) | | | | | 53 (8-114) | | 40 (7-90) |
| Working status | | | | | | | | |
| Baseline | 200 (89%) | | | | | 55 (87%) | | 255 (89%) |
| 6 weeks | 28 (15%) | | | | | 3 (5%) | | 31 (13%) |
| 3 months | 24 (12%) | | | | | 6 (11%) | | 30 (12%) |
| 6 months | 35 (17%) | | | | | 9 (15%) | | 44 (17%) |
| 1 year | 87 (42%) | | | | | 25 (42%) | | 112 (42%) |

| Median hospital costs | Central Hospitals (n = 198) | | | | | District Hospitals (n = 55) | | TOTAL |
|---|---|---|---|---|---|---|---|---|
| | Gustilo III (n = 106) | | | Gustilo I&II (n = 92) | | Gustilo III (n = 6) | Gustilo I &II (n = 49) | |
| | Nail (n = 44) | ExFix (n = 50) | Amputation (n = 12) | Nail (n = 66) | POP (n = 26) | POP (n = 6) | POP (n = 49) | |
| Length of stay | 15 (8–23) | 23 (13-39) | 29 (17-40) | 11 (8–17) | 7 (4–11) | 25 (13-44) | 9 (7–15) | |
| *Inpatient costs (2021 USD):* | | | | | | | | |
| Ward personnel | 115 | 154 | 141 | 75 | 52 | 23 | 9 | |
| Overhead | 6 | 10 | 12 | 5 | 3 | 15 | 5 | |
| Surgical implants | 150 | 11 | 0 | 150 | 0 | 0 | 0 | |
| Investigations | 36 | 36 | 18 | 36 | 36 | 17 | 17 | |
| Procedure personnel | 8 | 11 | 73 | 8 | 6 | 4 | 4 | |
| Procedure supplies | 39 | 47 | 44 | 30 | 26 | 53 | 47 | |
| Instruments | 1 | <1 | 28 | 1 | <1 | <1 | <1 | |
| Medication | 7 | 11 | 17 | 6 | 4 | 14 | 4 | |
| TOTAL | 348 (285-435) | 280 (191-427) | 333 (263-493) | 311 (287-358) | 127 (104-158) | 125 (97-172) | 86 (84-104) | |
| | | | | | | | | |
| *Outpatient costs:* | | | | | | | | |
| Clinic personnel | 6 | 6 | 6 | 6 | 6 | 1 | 1 | |
| Radiography | 54 | 54 | 0 | 54 | 54 | 17 | 17 | |
| Transportation | 21 | 36 | 18 | 22 | 17 | 15 | 31 | |
| Mean costs of re-admissions (SD) | 11 (31) | 29 (87) | 0 | 22 (71) | 51 (129) | 0 | 0 | |

*(Continued)*

**Table 1.** (Continued)

| Patient costs | Central Hospital (n = 224) | | | | | District hospital (n = 63) | | TOTAL |
|---|---|---|---|---|---|---|---|---|
| **Indirect costs** *(2021 USD)* | | | | | | | | |
| Lost productivity | 422 | 334 | 476 | 243 | 613 | 959 | 283 | |
| % of total societal cost | 46 | 44 | 49 | 36 | 63 | 81 | 64 | |
| TOTAL societal costs *(2021 USD)* | 916 (471−1,365) | 762 (504−1,318) | 966 (751−1,020) | 671 (291−1,380) | 970 (293−1,464) | 1,178 (312−1,867) | 445 (182-872) | |

From a societal perspective (healthcare and patient costs), the median empirical societal costs ranged from US$445 (IQR: 182−872) for plaster of Paris treatment (POP) in a district hospital for participants with a Gustilo I or II injury to US$1,178 (312−1,867) for POP treatment in a district hospital for a Gustilo III injury (See Table 1).

Regression modelling demonstrated substantial costs to patients in the year following their injury. Modelled household income was lower one year after injury (posterior mean: US$22.8; 95% CrIs: S$8.0-US$101.9) compared to pre-injury (posterior mean: US$75.8; 95% CrIs: US$18-US$569) (Fig 1). The posterior mean for patients reporting a monthly household income of USD$0 at one year post-injury was 20.5% (95% CrIs: 15.8-25.6%) compared to only 0.1% (95% CrIs: 0.0-0.5%) pre-injury. There were also substantial declines in the proportion of participants who reported being in work one year post-injury (posterior mean: 41%, 95% CrIs: 40.7- 42.5%) compared to being in work pre-injury (posterior mean: 88%, 95%CrIs: 87.5-88.7%) (Fig 2).

After exclusions of rare treatments, we included data from 253 participants who completed one-year follow-up, with data and inference from models summarised in Fig 3. The median posterior distribution for healthcare costs at one year ranged from US$112 (80% CrIs: US$84−148) for a patient with a POP in a district hospital with a Gustilo I/II injury to US$456 (80% CrIs: US$347−699) for a patient with a nail in a central hospital for a Gustilo III injury. Similarly, the median posterior distribution for societal costs at one year ranged from US$803 (80% CrI: US$-329–1,966) for a patient with a POP in a tertiary hospital with a Gustilo I/II injury to US$2,428 (80% CrIs: US$996 − 5,027) for a participant with a nail in a central hospital for a Gustilo III injury.

The smallest difference in cost-effectiveness from a societal perspective was between a POP in a district hospital and amputation for a participant with a Gustilo III injury with a posterior mean of US$-90 (95%HDI: −3,561–341) per QALY.

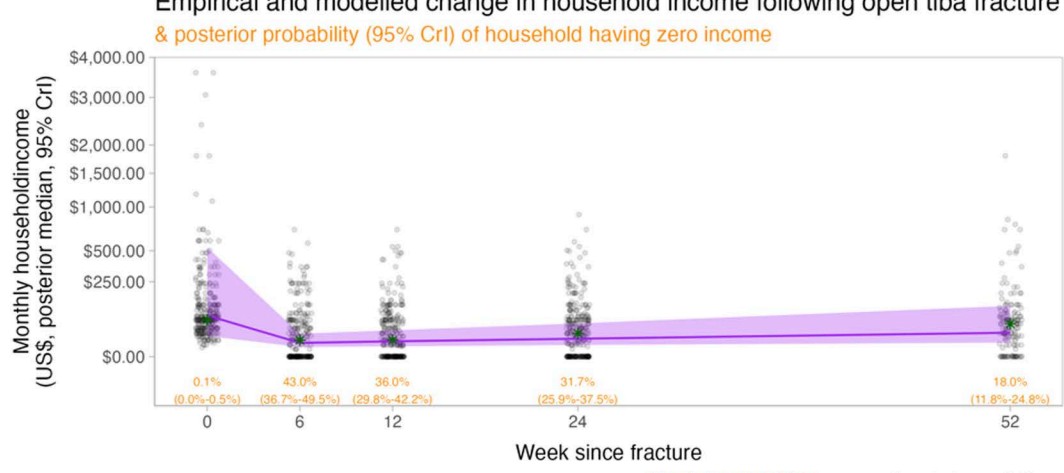

**Fig 1. Household income.**

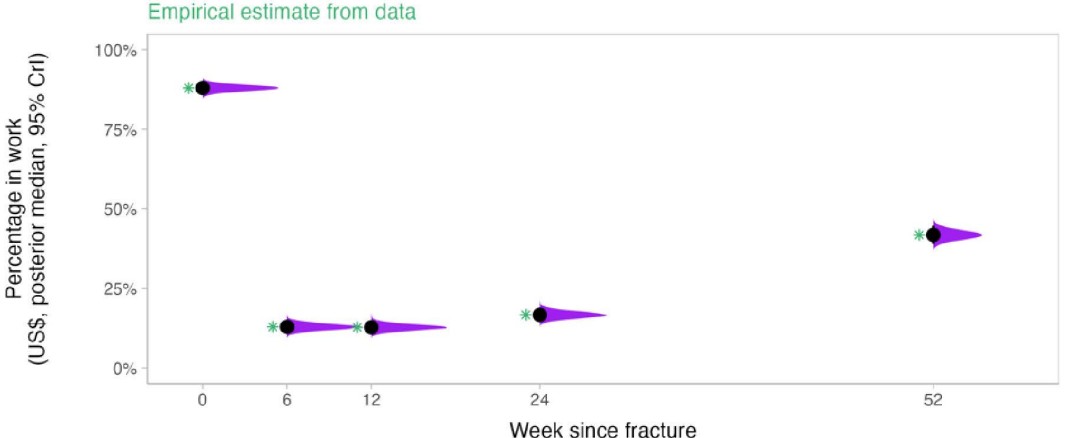

## Empirical and modelled percentage of participants in work
Empirical estimate from data

Purple histogram is overall posterior distribution

**Fig 2. Participants' working status.** (Green stars represent mean empirical percentage in work at each timepoint).

**Fig 3. Modelled healthcare and societal costs and EQ-5D index values for Gustilo I/II and Gustilo III for different orthopaedic treatments.**

The largest difference in cost-effectiveness from a societal perspective was between a nail and amputation for a Gustilo III injury with a posterior mean of US$2,290 (95%HDI: 36–4,547) per QALY (Table 2).

## Discussion

Our main finding was that open tibia fractures result in significant costs to patients, the healthcare system and society in Malawi. Orthopaedic interventions might not be affordable to the Malawi government, but substantial investment in trauma care will be required from government and stakeholders to reduce the large societal impact of injury [31]. Although the funding of orthopaedic treatment can be difficult in countries with very limited healthcare budgets, a re-balancing of health budgets (including from government and donors) is needed to prioritise trauma care to reduce the growing societal economic burden from injury.

We found that open tibia fractures are associated with very high societal costs ($904−3,405) irrespective of treatment modality. These high societal costs from open tibia fractures have also been reported from a systematic review in HICs (£27,123 to £50,197 [8]) and $2,560.81 to $2,664.59 in Tanzania [33]. Similarly, closed femoral shaft fractures in Malawi, had societal costs of $1,035 for early intramedullary nailing [34]. Furthermore, patients with open tibia fractures in Malawi who had an infection or Gustilo III injuries were associated with even higher healthcare costs which has also been reported in a cost-analysis of patient with open tibia fractures in Belgium [7]. The societal costs of open tibia fractures and other musculoskeletal injuries are higher than some infectious diseases in Malawi for example, *Enterobacterales* bloodstream infections: US$626.06 (SE 93.1) [35]). Funding for trauma care is lower than some other global health interventions, therefore, the prioritisation of healthcare interventions in global health does not match the societal costs [2]. More research needs to be conducted on the societal costs of different injuries to inform policymakers on prioritising healthcare budgets.

**Table 2. Modelled costs and cost-effectiveness of orthopaedic treatment strategies; Median posterior Healthcare and societal costs are reported in 2021 US$ (80% credible intervals); Mean posterior healthcare and societal costs per QALY are reported in US$ per QALY (95% Highest density intervals; POP=plaster of Paris, Differences comparisons are absolute differences.**

| Treatment or comparison | Healthcare cost (US$) | Societal costs (US$) | Healthcare cost per QALY (US$) | Societal cost per QALY (US$) |
| --- | --- | --- | --- | --- |
| **Gustilo I&II** | | | | |
| POP (district) | 115 (89-141) | 1,383 (874−1,910) | 505 (441-569) | 1 (−11−19) |
| POP (tertiary) | 208 (169-249) | 972 (210−1,743) | 719 (642-799) | 1,527 (482−2,566) |
| Intramedullary nailing | 406 (386-427) | 1,126 (700−1,548) | 932 (883-983) | 1,684 (872−2,499) |
| POP (district) vs POP (tertiary) | −169 (−225—115) | 404 (−565−1,408) | −214 (−284 - −144) | 375 (−827−1,577) |
| Intramedullary nailing vs POP (tertiary) | 184 (131-234) | 152 (798−1,098) | 213 (142-281) | 135 (−1023−1,274) |
| intramedullary nail vs POP (district) | 353 (312-395) | −251 (1,034-512) | 427 (371-484) | −240 (−1178−683) |
| **Gustilo III** | | | | |
| External Fixators | 386 (231-720) | 1,116 (−221−2,657) | 400 (347-452) | 933 (−1,328−3,145) |
| Intramedullary nailing | 456 (374-699) | 2,428 (996−5,027) | 451 (343-557) | 3,161 (738−5,636) |
| Amputation | 394 (218-642) | 1,182 (−237−2,701) | 373 (220-525) | 479 (−2,2718−3,686) |
| POP (district) | 164 (85-216) | 1,111 (−532−2,761) | 45 (−165−250) | −39 (−220−144) |
| POP (district) vs amputation | −279 (−479 - −76) | −26 (−3,090−2957) | −329 (−560 - −98) | −90 (−3,561 - −3,341) |
| External Fixator vs amputation | 18 (−102−140) | 17 (−1,780−1,836) | 26 (−116−171) | 43 (−2,071−2,196) |
| Intramedullary nailing vs amputation | 67 (−54−194) | 1,973 (20 −3 ,856) | 77 (−70−228) | 2,290 (36 −4 ,547) |
| External fixation vs POP (district) | 299 (115−476) | 89 (2,658−2,825) | 355 (151−561) | 133 (−2959−3,220) |
| Intramedullary nail vs POP (district) | 346 (164−524) | 2,000 (−693−4,683) | 407 (200−612) | 2,380 (−734−5,428) |
| Intramedullary nail vs external fixator | 49 (−34−130) | 1,942 (721−3,134) | 51 (−44−146) | 2,247 (819−3,673) |

Open tibia fractures have devastating economic consequences on patients and their households too [5]. Indeed, despite 89% of participants having been working pre-injury, only 42% reported working at one-year post-injury and 18% of households reported no income compared to 0.1% pre-injury. A prospective cohort study of patients with open tibia fractures in Uganda reported 100% of participants working prior to injury and only 20% working one year after injury, but this increased to 71% two years after injury [36] but may be less in Malawi. Qualitative interviews with patients with injuries in Ghana found that most financial losses for patients were due to loss of working wages rather than direct costs of medical treatment [37]. Indeed, patients with open tibia fractures in Malawi reported substantial loss of productivity wages between US$243 and US$959. This indirect cost from injury might be underestimated as the follow-up was one year and a substantial number of participants has not returned to work. Most participants were in the poorest poverty quintile and the injury drives households further into poverty and exacerbates inequalities further. The process of returning to work after sustaining severe open tibia fractures poses significant challenges for individuals in Malawi and more work needs to be done to protect against catastrophic financial expenditure [6]. A qualitative study exploring disability following an open tibia fracture in Malawi suggests that injury has a wide impact on the societal role of participants including family relocation, hunger and stigmatisation [5]. Further research should focus on factors and outcomes that improve the return of income such as social support and rehabilitation on injury [38].

As LMIC economies grow and motor transport becomes more common, fractures are placing a large financial strain on the economies [39]. The cost-effectiveness of orthopaedic treatment for patients with open tibia fractures was very high compared to non-operative management in district hospitals. Other orthopaedic treatments have been suggested to be cost-effective such as femoral shaft nailing compared to skeletal traction and the orthopaedic clinical officer programme compared to oral rehydration solution for diarrhoea or breast feeding promotion in Malawi [34,40,41]. Malawi is one of the poorest countries in the world with one of the lowest per capita expenditures on health in Africa [42]. Recent work in Malawi would suggest the willingness to pay threshold is very low which would suggest the Malawi government cannot afford many healthcare interventions including trauma care [31]. However, if the growing high societal costs from injury are to be reduced, investment in trauma care will need to be prioritised. This highlights the importance of government-donor collaboration and the need for more research on cost-effectiveness of trauma care.

There are several limitations of this study. Firstly, the payer's perspective includes government and internal donor costs. SIGN (Surgical Implant Generation Network) Fracture Care International currently donates intramedullary (IM) nails free of charge to many hospitals in LMICs, including Malawi [43]. This could lead to even greater cost savings for the Malawi health service than this study's estimates, as our analysis included the cost of the IM nail (US $150). Interventions are not always affordable and accessible in Malawi, where operative fracture treatment is not universally available in public hospitals. Future studies should include budget impact analyses assessing the affordability of adopting a new intervention from the payer's perspective [44]. Secondly, the limitations include that hospital costs were only estimated from Malawi and might not be generalisable to other LMICs which may include patient fees. Very few participants reported having health insurance, but our study was conducted in Government hospitals and therefore is not reflective of private hospitals in Malawi.

We did not consider the costs of building theatres or surgical training, which might increase the societal costs further. In Malawi, improved surgical training and expanding surgical capacity to manage the injury burden has been shown to reduce amputation rates [45]. The Lancet Commission on Global Surgery reported a global shortage of 1.1 million surgical and anaesthetic providers in 2015, most of whom are required in low- and middle-income countries (LMICs) [6]. The Commission's recommendation was to expand the surgical and anaesthetic workforce to at least a minimum density of 20 providers for every 100,000 population by 2030, as lower densities were correlated with increased maternal mortality [46]. Meeting this goal would involve enrolling an extra 1.27 million providers and a total investment of about $45 billion— an aspiration that is perhaps unattainable for most LMICs considering current resource limitations. Despite this, the high societal and economic price for treating injuries conservatively is too great. The WHO Economic Cost of Ill-Health Model

estimates that an economic loss from unattended surgical conditions of $20.7 trillion will occur in 128 countries from 2015 to 2030, with over half ($12.3 trillion) occurring in LMICs [3].

In conclusion, open tibia fractures result in significant societal costs in Malawi one year after injury both from the patients' and societal perspectives. Although the funding of orthopaedic treatment can be difficult in countries with very limited healthcare budgets, the costs to society of ignoring this issue are very high. A re-balancing of health budgets (including from government and donors) is needed to prioritise trauma care to reduce the growing societal economic burden from injury

## Supporting information

**S1 File. Topic guide for information collection.**
(DOCX)

**S2 File. Inclusivity in global research.**
(DOCX)

## Author contributions

**Conceptualization:** Alexander Thomas Schade, Linda Alinafe Sande, Maureen Sabawo, Leonard Banza Ngoie, Andrew John Metcalfe, David Griffith Lalloo, William James Harrison, Jason J Madan, Peter MacPherson.

**Data curation:** Alexander Thomas Schade, Nyamulani Nohakhelha, William James Harrison, Peter MacPherson.

**Formal analysis:** Alexander Thomas Schade, Ewan Tomeny, Jason J Madan, Peter MacPherson.

**Funding acquisition:** Alexander Thomas Schade, David Griffith Lalloo, William James Harrison, Peter MacPherson.

**Investigation:** Alexander Thomas Schade, Linda Alinafe Sande, William James Harrison, Peter MacPherson.

**Methodology:** Alexander Thomas Schade, Linda Alinafe Sande, Ewan Tomeny, Maureen Sabawo, Nyamulani Nohakhelha, Kaweme Mwafulirwa, Leonard Banza Ngoie, Andrew John Metcalfe, William James Harrison, Jason J Madan, Peter MacPherson.

**Project administration:** Alexander Thomas Schade, Maureen Sabawo, Nyamulani Nohakhelha, Kaweme Mwafulirwa, Andrew John Metcalfe, David Griffith Lalloo, William James Harrison, Peter MacPherson.

**Resources:** Alexander Thomas Schade, David Griffith Lalloo, Peter MacPherson.

**Software:** Alexander Thomas Schade, Peter MacPherson.

**Supervision:** Alexander Thomas Schade, Ewan Tomeny, Nyamulani Nohakhelha, Kaweme Mwafulirwa, Leonard Banza Ngoie, Andrew John Metcalfe, David Griffith Lalloo, William James Harrison, Jason J Madan, Peter MacPherson.

**Validation:** Alexander Thomas Schade, Peter MacPherson.

**Visualization:** Alexander Thomas Schade, Peter MacPherson.

**Writing – original draft:** Alexander Thomas Schade, Ewan Tomeny, Maureen Sabawo, William James Harrison, Jason J Madan, Peter MacPherson.

**Writing – review & editing:** Alexander Thomas Schade, Linda Alinafe Sande, Ewan Tomeny, Maureen Sabawo, Nyamulani Nohakhelha, Kaweme Mwafulirwa, Leonard Banza Ngoie, Andrew John Metcalfe, David Griffith Lalloo, William James Harrison, Jason J Madan, Peter MacPherson.

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
