## [Decision Letter · Decision Letter 0]

4 Apr 2025

Dear Dr. Schade,

Thank you for submitting your manuscript to PLOS ONE. After careful consideration, we feel that it has merit but does not fully meet PLOS ONE’s publication criteria as it currently stands. Therefore, we invite you to submit a revised version of the manuscript that addresses the points raised during the review process.

The study suggests that intramedullary nailing is the most cost-effective treatment for open tibia fractures in Malawi. However, the practical implementation of this finding might require further discussion:

- Training of orthopedic surgeons: Given the shortage of trained orthopedic surgeons in Malawi, the feasibility of expanding surgical treatments should be addressed. If intramedullary nailing is to be widely adopted, investment in surgeon training and infrastructure is necessary.

- Accessibility of surgical care: Since most fractures in Malawi are treated non-operatively by Clinical Officers, the study should discuss whether a shift toward surgical management is realistically achievable within the current healthcare system.

These aspects would strengthen the clinical relevance of the study and help translate findings into practical recommendations.

We look forward to receiving your revised manuscript.

Kind regards,

Xiaoen Wei

Academic Editor

PLOS ONE

Journal Requirements:

4. Thank you for stating the following in your Competing Interests section: “no. competing interests.”

5. Thank you for uploading your study's underlying data set. Unfortunately, the repository you have noted in your Data Availability statement does not qualify as an acceptable data repository according to PLOS's standards. At this time, please upload the minimal data set necessary to replicate your study's findings to a stable, public repository (such as figshare or Dryad) and provide us with the relevant URLs, DOIs, or accession numbers that may be used to access these data. For a list of recommended repositories and additional information on PLOS standards for data deposition, please see https://journals.plos.org/plosone/s/recommended-repositories .

6. Please note that in order to use the direct billing option the corresponding author must be affiliated with the chosen institute. Please either amend your manuscript to change the affiliation or corresponding author, or email us at plosone@plos.org with a request to remove this option.

Additional Editor Comments:

The study suggests that intramedullary nailing is the most cost-effective treatment for open tibia fractures in Malawi. However, the practical implementation of this finding might require further discussion:

Given the shortage of trained orthopedic surgeons in Malawi, the feasibility of expanding surgical treatments should be addressed. If intramedullary nailing is to be widely adopted, investment in surgeon training and infrastructure is necessary.

Since most fractures in Malawi are treated non-operatively by Clinical Officers, the study should discuss whether a shift toward surgical management is realistically achievable within the current healthcare system.

These aspects would strengthen the clinical relevance of the study and help translate findings into practical recommendations.

Reviewers' comments:

Reviewer's Responses to Questions

**Comments to the Author**

1. Is the manuscript technically sound, and do the data support the conclusions?

Reviewer #1: Yes

2. Has the statistical analysis been performed appropriately and rigorously?

Reviewer #1: Yes

3. Have the authors made all data underlying the findings in their manuscript fully available?

Reviewer #1: Yes

4. Is the manuscript presented in an intelligible fashion and written in standard English?

Reviewer #1: Yes

Reviewer #1: The study suggests that intramedullary nailing is the most cost-effective treatment for open tibia fractures in Malawi. However, the practical implementation of this finding might require further discussion:

- Training of orthopedic surgeons: Given the shortage of trained orthopedic surgeons in Malawi, the feasibility of expanding surgical treatments should be addressed. If intramedullary nailing is to be widely adopted, investment in surgeon training and infrastructure is necessary.

- Accessibility of surgical care: Since most fractures in Malawi are treated non-operatively by Clinical Officers, the study should discuss whether a shift toward surgical management is realistically achievable within the current healthcare system.

These aspects would strengthen the clinical relevance of the study and help translate findings into practical recommendations.

**Do you want your identity to be public for this peer review?** For information about this choice, including consent withdrawal, please see our Privacy Policy

Reviewer #1: **Yes: ** Gianmarco Vavalle

---

## [Author Response · Author response to Decision Letter 1]

31 May 2025

I write on behalf of my co-authors to thank the reviewers for their careful review and appreciate the opportunity to respond with a revised submission.

The authors have declared that no competing interests exist.

Please see the point-by-point responses below.

---

## [Decision Letter · Decision Letter 1]

21 Jun 2025

Dear Dr. Schade,

We look forward to receiving your revised manuscript.

Kind regards,

Xiaoen Wei

Academic Editor

PLOS ONE

Reviewers' comments:

Reviewer's Responses to Questions

**Comments to the Author**

Reviewer #2: (No Response)

Reviewer #3: (No Response)

2. Is the manuscript technically sound, and do the data support the conclusions?

Reviewer #2: Yes

Reviewer #3: Partly

3. Has the statistical analysis been performed appropriately and rigorously?

Reviewer #2: Yes

Reviewer #3: Yes

4. Have the authors made all data underlying the findings in their manuscript fully available?

Reviewer #2: Yes

Reviewer #3: (No Response)

5. Is the manuscript presented in an intelligible fashion and written in standard English?

Reviewer #2: Yes

Reviewer #3: Yes

**Reviewer #2: ** While I give kudos to the authors, I think the discussion could be more robust judging by data analysis provided. So, I suggest the author beef up the discussion section with relevant citations. This will justify the data analysis provided.

**Reviewer #3:**  Abstract

- Open tibia fracture can have an equal impact on women, what makes it more detrimental to men according to your research?

- We obtained quality adjusted life years (QALYs) from the EuroQoL 5 Dimension 3 Level (EQ-5D-3L)…

- Why were costs reported in 2021; at least inflate to 2024, or better still, report using 2024 costs.

Background

Method

- The study protocol was published -------include date it was published. Ln 99

- Write the statement to make meaning or remove the bracket and let it be a complete sentence: At recruitment …. study. Lns 133-135

- Each procedure was observed between 1-3 times. - What was observed? Ln 154

- Did the difference in the way the costs were collected impact the study?

- How was missing data inputted?

Result

- What was the incremental change in Costs and QALYs. Can the QALY per treatment type and location (central vs district) be included? What is the difference between costs and QALYs based on where the treatment took place?

- Can you include a table showing the costs, QALYs, Incremental Costs, Incremental QALYs and ICER values

-

**Do you want your identity to be public for this peer review?** For information about this choice, including consent withdrawal, please see our Privacy Policy

Reviewer #2: **Yes: ** Anthony Olasinde

Reviewer #3: No

---

## [Author Response · Author response to Decision Letter 2]

17 Jul 2025

1. Reviewer 2

Comment #1:

Reviewer #2: While I give kudos to the authors, I think the discussion could be more robust judging by data analysis provided. So, I suggest the author beef up the discussion section with relevant citations. This will justify the data analysis provided.

Author’s Response: We thank the reviewer for their positive feedback and constructive suggestion. In response, we have substantially revised and expanded the Discussion section to better contextualise our findings and strengthen the link between the data analysis and its implications. Specifically, we have incorporated additional references to relevant literature (1-4), including recent studies in similar settings, to support our interpretations and highlight the significance of our results.

Lines: 275

“Orthopaedic interventions might not be affordable to the Malawi government, but substantial investment in trauma care will be required from government and stakeholders to reduce the large societal impact of injury (1)”

Line 296

Open tibia fractures have devastating economic consequences on patients and their households too (2).

Line 313

Further research should focus on factors and outcomes that improve the return of income such as social support and rehabilitation on injury (3).

Line 315

As LMIC economies grow and motor transport becomes more common, fractures are placing a large financial strain on the economies (4).

”

Reviewer #3:

Comment #1:

Open tibia fracture can have an equal impact on women, what makes it more detrimental to men according to your research?

Author’s Response: Thank you for your comment. Open tibia fractures can indeed have a significant impact on both men and women. However, in our cohort, 84% of patients were male, which is consistent with global data showing that traumatic injuries disproportionately affect young men (4). In the Malawian context, cultural norms often position men as the primary income earners (5). As a result, the economic consequences of injury may be more pronounced for men, particularly in terms of lost income and reduced household stability. To our knowledge, this socioeconomic impact has not been previously documented in this setting and warrants further investigation.

Comment #2:

We obtained quality adjusted life years (QALYs) from the EuroQoL 5 Dimension 3 Level (EQ-5D-3L)…

Author’s Response: Many thanks, we have amended the abstract accordingly.

Comment #2:

Why were costs reported in 2021; at least inflate to 2024, or better still, report using 2024 costs.

Author’s Response: Thank you for your comment. As outlined in the methods section, recruitment took place between 12 February 2021 and 15 March 2022. To reflect the timing of data collection and ensure consistency, costs were reported in 2021 values.

Comment #3:

The study protocol was published -------include date it was published. Ln 99

Author’s Response: Thank you for highlighting this. The study protocol was published in 2021, and we have now included the publication year at line 99 in the Methods section for clarity.

Lines: 100

“The study protocol was previously published in 2021”

Comment #3:

Write the statement to make meaning or remove the bracket and let it be a complete sentence: At recruitment …. study. Lns 133-135

Author’s Response: Thank you for your suggestion. We have revised and completed the sentence for clarity.

Line 134

“At recruitment, participants were asked to retrospectively complete the questionnaire for their health state pre-injury and at each subsequent follow-up visit, study.”

Comment #3:

Each procedure was observed between 1-3 times. - What was observed? Ln 154

Author’s Response: Thank you for highlighting this point. We have revised the manuscript to clarify what aspects of the procedure were observed. Specifically, we have added a description of the patient treatment pathway in the main text and detailed which procedures were directly observed in Appendix A. This should make the scope and focus of the observations clearer to the reader.

Comment #3:

- Did the difference in the way the costs were collected impact the study?

Author’s Response: Thank you for your comment. Due to the lower frequency of open tibia fractures in district hospitals, we were unable to conduct time-and-motion analyses at those sites. This difference in data collection methods may have introduced some limitations, which we have now acknowledged and discussed in the limitations section of the manuscript.

Comment #4

- How was missing data inputted?

Thank you for your question. Observations with missing data were excluded from the relevant analyses. Given the small number of missing values, we did not attempt formal imputation. Where appropriate, we conducted sensitivity analyses to assess the potential impact of missing data on our findings.

Comment #5

- What was the incremental change in Costs and QALYs. Can the QALY per treatment type and location (central vs district) be included? What is the difference between costs and QALYs based on where the treatment took place?

- Can you include a table showing the costs, QALYs, Incremental Costs, Incremental QALYs and ICER values

Thank you for your helpful comments. As clarified in the revised manuscript, Table 2 presents the total costs, incremental costs, and cost per QALY (ICER) for each treatment type, stratified by treatment location (central vs district hospitals). This allows for direct comparison of cost-effectiveness across settings. QALYs and incremental QALYs were reported in our previous publication (7) and are not reproduced in full here to avoid duplication. However, the ICER values in Table 2 incorporate these published QALY estimates to provide a comprehensive assessment of cost-utility.

---

## [Decision Letter · Decision Letter 2]

19 Aug 2025

Economic burden and cost-effectiveness of treatments for open tibia fractures in Malawi: economic analysis of a multicentre prospective cohort study

PONE-D-24-40054R2

Dear Dr. Schade,

We’re pleased to inform you that your manuscript has been judged scientifically suitable for publication and will be formally accepted for publication once it meets all outstanding technical requirements.

Kind regards,

Xiaoen Wei

Academic Editor

PLOS ONE

Additional Editor Comments (optional):

Reviewers' comments:

Reviewer's Responses to Questions

**Comments to the Author**

Reviewer #2: All comments have been addressed

2. Is the manuscript technically sound, and do the data support the conclusions?

Reviewer #2: Yes

3. Has the statistical analysis been performed appropriately and rigorously?

Reviewer #2: Yes

4. Have the authors made all data underlying the findings in their manuscript fully available?

Reviewer #2: Yes

5. Is the manuscript presented in an intelligible fashion and written in standard English?

Reviewer #2: Yes

Reviewer #2: I have no further coment to make. The authors have addressed the previously ones. THE Editorial should make the final decision on the manuscript

**Do you want your identity to be public for this peer review?** For information about this choice, including consent withdrawal, please see our Privacy Policy

Reviewer #2: **Yes: ** Anthony Olasinde

---

## [Editor Report · Acceptance letter]

PONE-D-24-40054R2

PLOS ONE

Dear Dr. Schade,

I'm pleased to inform you that your manuscript has been deemed suitable for publication in PLOS ONE. Congratulations! Your manuscript is now being handed over to our production team.

Kind regards,

on behalf of

Dr. Xiaoen Wei

Academic Editor

PLOS ONE